# Microstructure and Wear Resistance of Mg₂Si–Al Composites Fabricated Using Semi-Solid Extrusion

**Xiaobo Liu** [1,*], **Miao Yang** [2,*], **Dekun Zhou** [1] and **Yuguang Zhao** [3]

1    College of Mechanical Engineering, Beihua University, Jilin 132021, China; zhoudekun2000@163.com
2    Engineering Training Center, Beihua University, Jilin 132021, China
3    College of Materials Science and Engineering, Jilin University, Changchun 130022, China; zhaoyg@jlu.edu.cn
*    Correspondence: stone-666@126.com (X.L.); yangmiao1021@163.com (M.Y.);
      Tel.: +86-186-0449-7783 (X.L.); +86-186-0449-8817 (M.Y.)

**Abstract:** In situ Mg₂Si–Al composites were prepared by using gravity casting and semi-solid extrusion. After P modification, the primary Mg₂Si transformed to polygonal blocks. Extraction tests showed that the Mg₂Si crystals had octahedral and tetrakaidekahedral morphologies. The semi-solid microstructure of the double-spheroidized α-Al matrix and reinforced-phase Mg₂Si was successfully obtained by using semi-solid extrusion. Extraction tests showed that the Mg₂Si crystals had a spherical morphology. Dry sliding wear behaviors of in situ Mg₂Si–Al composites fabricated by using gravity casting and semi-solid extrusion with isothermal heat treatment holding times of 50, 60, and 160 min against 45 steel, under conditions of different sliding speeds and loads, were investigated. The worn surfaces were analyzed using SEM and EDS techniques. The results showed that Mg₂Si–Al composites fabricated by using semi-solid extrusion were superior in terms of wear resistance to Mg₂Si–Al composites fabricated by using gravity casting, because the former had uniformly distributed spherical reinforced phase particles of Mg₂Si with weaker stress concentration around the particles, delaying the generation and expansion of cracks. The Mg₂Si particles were not easily detached from the matrix, and once they fell off, the Mg₂Si particles only served as spherical abrasive grains, with relatively small cutting and wear properties for the composite material. It was found that the Mg₂Si/Al composite fabricated by using semi-solid extrusion with an isothermal heat treatment holding time of 60 min had the best wear resistance. The failure mechanisms of Mg₂Si/Al composites were found to be mainly adhesive wear and abrasive wear.

**Keywords:** in situ Mg₂Si–Al composite; semi-solid extrusion; microstructure; wear resistance

## 1. Introduction

Research into aluminum-based composite materials has received increasing attention from the materials and engineering community in recent years. It is known that Mg₂Si intermetallic compounds are easily formed during the aluminum alloy casting process; this is a thermodynamically stable phase formed in situ during solidification. Mg₂Si is very suitable as a reinforcement for aluminum matrix composites, as Mg₂Si has high hardness of $4.5 \times 10^9$ $Nm^{-2}$, a high melting temperature of 1085 °C, a high elastic modulus of 120 GPa, a low density of $1.99 \times 10^3$ kg m⁻³, and a low CTE (coefficient of thermal expansion) of $7.5 \times 10^{-6}$ K⁻¹ [1,2]. It shows a clean interface, good compatibility, and high bonding strength with the matrix. In addition, Mg₂Si–Al composites are far superior in terms of machinability and formability compared to ceramic particle–reinforced aluminum composites. Therefore, Mg₂Si/Al composite materials have great market potential and broad application prospects as a manufacturing material for lightweight components such as automobile piston cylinder liners. However, the primary Mg₂Si phases obtained by the ordinary smelting method are coarse dendrites with a poor microstructure

and distribution, which split the matrix resulting in poor strength, poor plasticity, and insufficient wear resistance of the $Mg_2Si$–Al composite. Therefore, changing the size, morphology, and distribution of the coarse dendritic $Mg_2Si$, thereby improving its comprehensive performance, has become the primary problem to be solved. Researchers [3–7] have performed much work on preparation methods, microstructure control, and material properties, and have made some progress, but further research is still needed.

Metal semi-solid processing technology, which is a more effective method for changing and controlling the microstructure of composite materials, is combined with the preparation process of the composite material to change its microstructure characteristics so that the composite matrix is non-dendritic and the reinforcement can be redistributed and homogenized. Compared with traditional casting and forging processes, semi-solid forming has a number of advantages, such as a low forming temperature, low deformation resistance, macro-segregation, and less solidification shrinkage [8]. One feature is that the microstructure of the material changes from dendrites to non-dendritic or globular crystals, resulting in a significant increase in strength and toughness. The preparation methods of semi-solid alloy billets mainly include the electromagnetic stirring method [9], the spray deposition method [10], the mechanical stirring method [11], the strain-induced melt activation method [12], the isothermal heat treatment method [13], and the cooling slope cast method [14]. It is worth mentioning in particular that by using the isothermal heat treatment method, ideal spherical grains can be obtained by a one-step heat treatment. This method used to be mainly applied to ZA (high aluminum zinc based alloy) alloys and has fewer applications in aluminum alloys and aluminum matrix composites. For the $Mg_2Si$–Al composite, a single one-step heat treatment process can doubly spheroidize the structure, which can improve the performance of the composite. In a study of the semi-solid microstructure, Canyook et al. [15] studied the evolution of the semi-solid microstructure of 365 aluminum alloys. Qin et al. [13,14] prepared semi-solid microstructures of $Mg_2Si$–Al composites by the strain induction method, the isothermal heat treatment method, and the cooling slope cast method, and studied the evolution law and mechanism of the semi-solid microstructure. In the study of wear resistance, some research has been done on the wear resistance of $Mg_2Si$–Al composites by using gravity casting, but research on the wear resistance of $Mg_2Si$–Al composites by using semi-solid extrusion is less common. Ahlatci [16] studied the effect of Mg on the friction and wear behavior of Al-12Si alloys. It was found that the particle size and particle number of the $Mg_2Si$ primary phase increased, and the hardness and wear resistance of the material increased with the addition of Mg. Wu et al. [17] studied the effect of Bi on the dry sliding friction and the wear behavior of Al–15%$Mg_2Si$ composites. It was found that after the addition of 0.2–8.0% Bi, the wear rate and friction coefficient of the Al–15%$Mg_2Si$ composites were both reduced due to the deterioration and self-lubrication of Bi. The wear mechanism changed from the mixed form of peeling wear, adhesive wear, and abrasive wear to adhesive wear and abrasive wear, indicating that the addition of Bi was beneficial to the improvement of the wear resistance of the composite. Liu et al. [18] studied the resistance of Al–20$Mg_2Si$–10Si composites prepared by gravity casting, squeeze casting, and semi-solid extrusion. They found that the wear rate of composites fabricated by using semi-solid extrusion was minimal under same load and wear particle size. Nadim et al. [19] studied the dry sliding wear of Al–15$Mg_2Si$–2Fe composites and found that Fe had negative effects on the sliding wear resistance of Al–15$Mg_2Si$–2Fe composites because of surface/subsurface micro-cracking and an unstable tribolayer, which were caused by β–Al5FeSi compounds.

In the present study, in situ $Mg_2Si$–Al composites were prepared by using gravity casting and semi-solid extrusion. The small block-shaped primary $Mg_2Si$ phases were obtained by gravity casting, while the spherical reinforced $Mg_2Si$ phases and matrix structure were successfully obtained by isothermal heat treatment technology. The wear resistances of $Mg_2Si$–Al composites were studied. The aim was to develop a new way to produce high-performance $Mg_2Si$–Al composites, with the hope of promoting the further development of automotive wear-resistant composites.

## 2. Materials and Methods

### 2.1. Material Preparation

Commercial Al–20Si master alloy (ingot) and magnesium (ingot, >98.0% purity) were used as starting materials. About 700 g of Al–20Si alloy was melted in a graphite crucible, which was heated to 200 °C in an electric resistance furnace. After the alloy was completely melted, it was held for 5 min. About 103 g of 300 °C pre-heated magnesium coated with aluminum foil was added into the Al–Si melt at 690–710 °C. The designed content of $Mg_2Si$ was 20 wt.%. A total of 0.5 wt.% of P via a Cu–14 wt.% P master alloy was added into the melts at 750 °C. The mass of Cu–14 wt.% P master alloy was about 29 g. After holding for 20 min, the composite melts were poured into steel die to produce ingots of φ54 mm × 90 mm (φ represents the diameter of the sample). The ingots were then machined to a size of φ54 mm × 60 mm. They were heated in an electric resistance furnace and held for 50, 60, 100, and 160 min at 565 °C, respectively. Subsequently, each set of the sample was placed in a preheated extrusion die and extruded into a sample of φ66 mm at 255 MPa for 35 s.

### 2.2. Microstructure Analysis

The specimens were characterized using an X-ray diffraction (XRD) (D/Max 2500 PC Rigaku, Tokyo, Japan) machine, using a Cu–Kα radiation target with a working current of 300 mA and a working voltage of 50 kV. Metallographic specimens were polished through standard methods using a 0.5 vol% HF aqueous solution and examined using an OLYMP USBX-60 metallographic microscope (Olympus Corporation Co. Ltd, Tokyo, Japan). The microstructure and wear surface of the sample were observed using a JSM-5310 scanning electron microscope (SEM) (JSM-5310, Japan Electron Optics Laboratory Co. Ltd, Tokyo, Japan) with energy-dispersive spectroscopy (EDS) (Link-Isis, Oxford, UK). The structural characteristics of the crystal and its crystal defects were observed using a Model JSM-6700F field (Japan Electron Optics Laboratory Co. Ltd, Tokyo, Japan) emission scanning electron microscope.

### 2.3. Hardness Tests

The hardness value was determined from the average value of seven hardness readings on each sample with a hardness tester (Brinell Hardness Tester HB-3000B, Shanghai Chiasson Instrument Co. Ltd, Shanghai, China), using a load of 7350 N and a holding time of 30 s.

### 2.4. Wear Tests

The dry sliding friction wear test was carried out using an MG-200 type friction and wear tester with a disc-shaped material of 45 steel. The pin-on-disk wear test was used. The sample size was φ5 mm × 12 mm, with a surface roughness of 0.9 μm. The samples were washed twice with alcohol before and after the test, and the mass of the samples was weighed with an electronic balance (Sartorius Genius ME215P, Sartorius Co. Ltd, Gottingen, Germany) with a precision of 100,000, before and after the test. The density of the worn specimen was measured by the Archimedes method, and the calculation formula was as shown in Equation (1). Finally, the wear loss was divided by the sample density to obtain the wear volume.

$$\rho = \frac{m}{m - m_{\mathrm{w}}} \tag{1}$$

where $m$ is the mass of the sample in air, and $m_{\mathrm{w}}$ is the mass of the sample in distilled water.

The specific parameters of the dry sliding wear test are shown in Table 1. The wear distance was 629 m under both variable loads and variable speed conditions. Three specimens were tested in each condition, and the wear volume was determined from the average value of them.

**Table 1.** Experimental parameters for Test I and II.

| Experiments | Velocity (r/min) | Load (N) |
|:---:|:---:|:---:|
| Test I | 700 | 20 |
| | | 30 |
| | | 40 |
| | | 50 |
| Test II | 300 | 30 |
| | 700 | |
| | 1000 | |

## 3. Results and Discussion

### 3.1. As-Cast Microstructure of the Composite

In the experiment, the alloy components were all hypereutectic, so in the $Mg_2Si$–Al solidification process, $Mg_2Si$ particles first appeared, and then Al and $Mg_2Si$ solidified in a eutectic form. The composition of the composite in this experiment was Al–20$Mg_2Si$–10Si. According to previous research [18], the solidification process of $Mg_2Si$–Al composites is as follows:

$$L \rightarrow L_1 + Mg_2Si_P \rightarrow L_2 + (Al + Mg_2Si)e + Mg_2Si_P \rightarrow (Al + Si + Mg_2Si)_e + (Al + Mg_2Si)_e + Mg_2Si_P \quad (2)$$

where the subscript P represents the primary phase and e represents the eutectic phase. Because of non-equilibrium in the solidification process, other phases, such as the eutectic Si phase, appeared.

Figure 1 shows the as-cast microstructure of the $Mg_2Si$–Al composites modified by phosphorus, fabricated by using gravity casting and semi-solid extrusion with different isothermal heat treatment holding times. It can be seen that the $Mg_2Si$ reinforcing phase and the $\alpha$-Al matrix were spheroidized after isothermal heat treatment for 50 min, 60 min, and 160 min, as compared with ordinary gravity casting. The morphology of the primary $Mg_2Si$ phase of gravity-cast $Mg_2Si$–Al composites was polygonal or quadrilateral, with an average size of about 35 μm, and the $\alpha$-Al phase was dendritic, as shown in Figure 1a. When the holding time was 50 min, the rose-like characteristics of $\alpha$-Al phase disappeared almost entirely, showing obvious spherical or ellipsoidal shapes with different particle sizes and uneven distribution. At the same time, the $Mg_2Si$ reinforced phase was also converted into spherical particles, but the roundness and size were not uniform, and the size was only 20 μm in part, while the other part was increased to 50 μm, as shown in Figure 1b. As the holding time increased to 60 min, the $\alpha$-Al phase was spheroidized and the size was relatively uniform, showing a regular spherical or ellipsoidal shape, but its size reached 85 μm, an increase compared to at 50 min. $Mg_2Si$ exhibited a fine spherical shape which was more rounded than that at 50 min, and the individual dimensions were increased to 60 μm, as shown in Figure 1c. As the holding time was further extended to 160 min, the spherical $\alpha$-Al increased to 120 μm and the morphology gradually became non-rounded, while the primary $Mg_2Si$ phase became more rounded and uniform in size but remained spherical. The size was about 30 μm, as shown in Figure 1d. In addition, the eutectic structure in the semi-solid structure was refined compared to the composite material prepared by gravity casting. When the temperature was maintained for 50 min, the eutectic structure was relatively small, and was more refined as the holding time was extended to 60 min, while the eutectic structure was coarsened when the holding time was extended to 160 min.

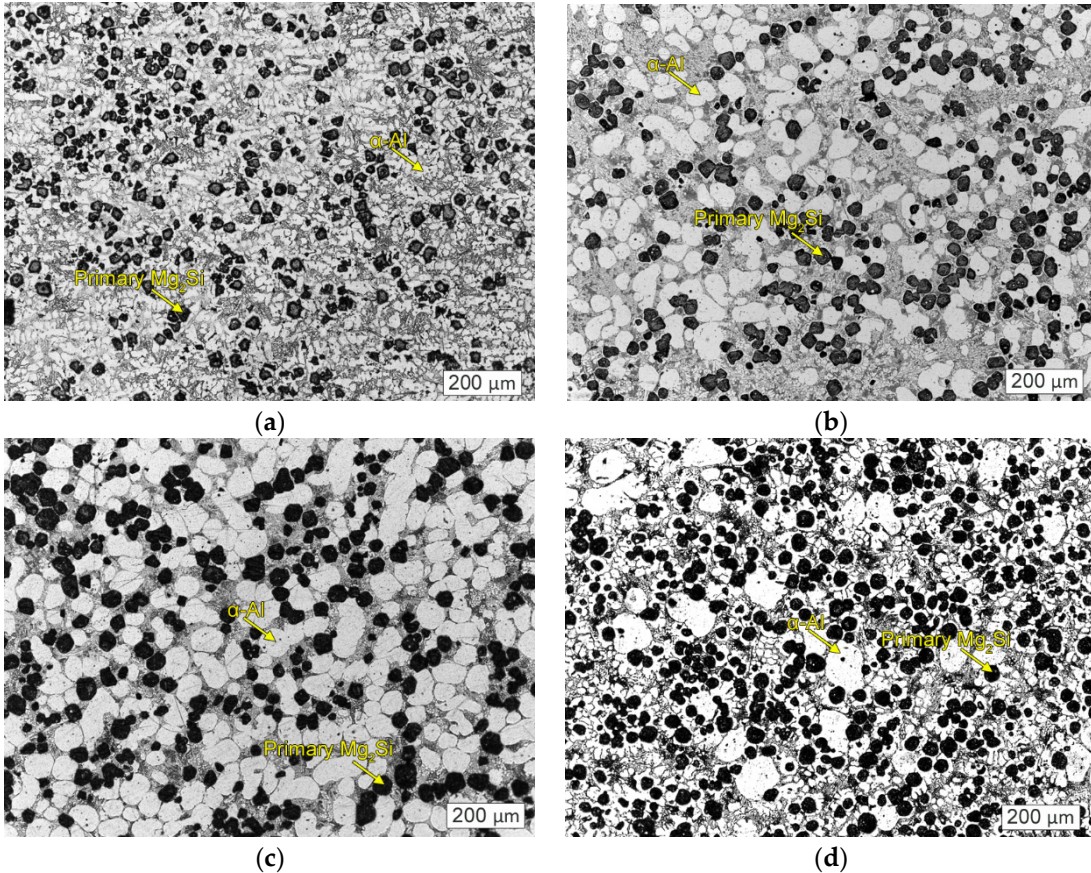

(a)

(b)

(c)

(d)

**Figure 1.** Microstructures of Mg$_2$Si–Al composites fabricated by using (**a**) gravity casting and semi-solid extrusion with isothermal heat treatment holding times of (**b**) 50 min, (**c**) 60 min, and (**d**) 160 min.

Figure 2 shows the XRD patterns of Mg$_2$Si–Al composites. Both of the Mg$_2$Si–Al composites fabricated by using gravity casting and semi-solid extrusion were composed of Al, Mg$_2$Si, CuAl$_2$, and Si phases. It was found that the morphologies of their corresponding phases were different, but the composition phases were the same.

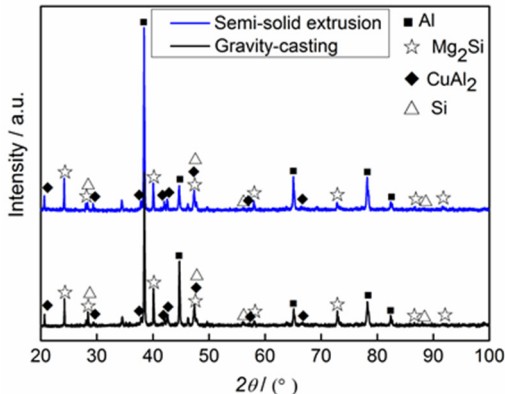

**Figure 2.** XRD spectra of Mg$_2$Si–Al composites.

Figure 3a,b show the octahedral morphology and tetrakaidecahedral morphology of Mg$_2$Si crystals extracted from Mg$_2$Si–Al composites fabricated by using gravity casting after P metamorphism. Figure 3c shows spherical morphology of an Mg$_2$Si crystal extracted from Mg$_2$Si–Al composite prepared by semi-solid extrusion, with an isothermal heat treatment holding time of 160 min. It shows that the

Mg$_2$Si particles became rounded and had a regular spherical shape when the isothermal heat treatment holding time was increased to 160 min.

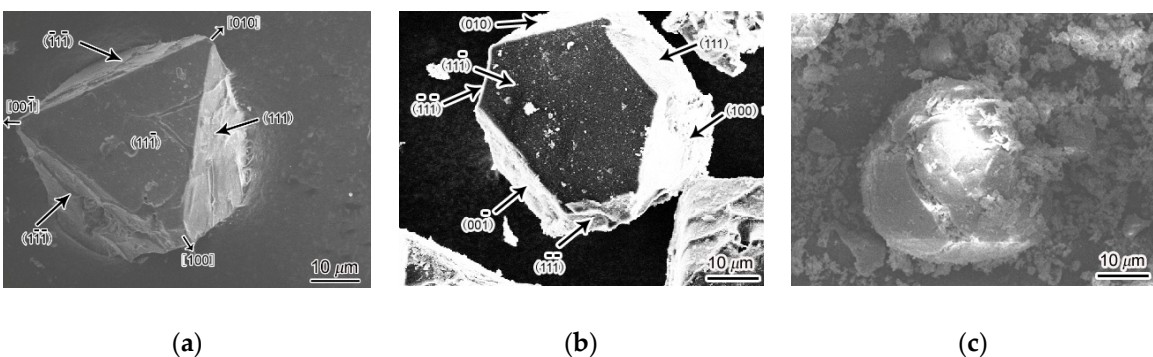

**(a)**　　　　　　　　　　　　**(b)**　　　　　　　　　　　　**(c)**

**Figure 3.** (**a**) Octahedral morphology and (**b**) tetrakaidecahedral morphology of Mg$_2$Si crystals extracted from Mg$_2$Si–Al composites prepared by gravity casting and (**c**) spherical morphology of an Mg$_2$Si crystal extracted from Mg$_2$Si–Al composite prepared by semi-solid extrusion with an isothermal heat treatment holding time of 160 min.

During solidification of the alloy, the primary Mg$_2$Si usually appeared as coarse dendrites without the corresponding microstructure control (such as modification treatment). These were actually formed by connecting Mg$_2$Si octahedral crystals together due to instability of the interface. In this study, Cu–14 wt.% P master alloy was used as a modifying agent of the Mg$_2$Si–Al composites. After P modification, the primary M$_{g2}$Si transformed from dendrite to a polygonal block, as shown in Figure 1a. Because Mg$_2$Si has a face-centered cubic crystal structure and the growth mode is typical of facet growth, under ideal conditions, Mg$_2$Si will grow into an octahedron, as shown in Figure 3a. Its growth surface was composed of densely packed crystal planes (111), and the growth direction was <100>. In addition, adding phosphorus formed a heterogeneous core Mg$_3$(PO$_4$)$_2$ compound [20], which was located at the center of the Mg$_2$Si crystal. The existence of heterogeneous cores increased the number of crystalline cores and generated more crystal grains, which caused the supersaturation of solutes to increase rapidly during the crystal growth process. As a result, the growth velocity of preferentially grown <100> crystal orientation decreased, so that the <100> crystal orientation changed to the {100} plane, thereby forming a tetrahedron as shown in Figure 3b. This is consistent with the research results of Qin et al. [21].

After semi-solid isothermal heat treatment, the morphology of the enhanced phase Mg$_2$Si changed from a polygonal block to a sphere, as shown in Figures 1b–d and 3c. Studies have shown that, from the perspective of thermodynamics, the spheroidization of grains is a spontaneous process during the heat treatment process. However, in the actual heat treatment process, even after a reasonable isothermal heat treatment, an ideal perfect sphere cannot be achieved. Mg$_2$Si is the primary phase of the alloy, has a higher melting point, and the spheroidization process is relatively simple; that is, sharp corners are dissolved and passivated to become round. With the extension of the holding time, the diffusion of most of the Mg$_2$Si phase and the liquid phase slightly reduced the size of the crystal grains, while a few of them increased in size due to coarsening and maturation.

*3.2. Wear Resistance*

The friction and wear curves of Mg$_2$Si–Al composites under different loads at a fixed sliding rate of 700 r/min are shown in Figure 4. The curves show that the wear volume gradually increased as the applied load increased when the sliding rate was constant. According to the change law of wear volume, the relationship between wear volume and load can be divided into three areas: the low-load area (20–30 N), medium-load area (30–40 N), and high-load area (40–50 N). In the low-load region, which is called the mild wear state [22], the wear volume showed a slow linear increasing trend as the load increased. In the medium-load area, the wear volume increased non-linearly with the increase of

the load, while in the high-load area, the increase in the wear volume was gentle. These two stages are called severe wear conditions [22]. It was also found in the experiment that a more severe vibration was generated during the high loading stage. In the low-load area, alumina was generated due to frictional heat generation on the wear surface. This oxide film was very hard and sat between the abrasive disc and the wear surface, so it had a very large protective effect on the wear surface. After being worn away, this layer of oxide film could be quickly formed again, thus reducing the wear on the material surface of the grinding disc and delaying the wear failure process. At the same time, work hardening also played a certain protective role. The wear method was relatively simple; there was only slight adhesive wear and slight cutting of the wear surface by hard points, so the wear surface peeled off slowly, resulting in less increase in wear volume. In the medium-load area, as the load increased, the wear increased, and the peeling rate of the wear surface was much faster than the rate of oxide film formation and work hardening, so the wear volume increased rapidly. The primary $Mg_2Si$ particles had high hardness and excellent abrasion resistance. In this experiment, the $Mg_2Si$ particles were small in size and dispersed in the soft aluminum matrix, forming a good combination of hard points and soft matrix, which improved the carrying capacity of $Mg_2Si$–Al composites. In the high-load area, the small $Mg_2Si$ particles were prominent, and their anti-wear effect gradually became prominent, which caused the increase in the wear volume of the composite material to slow down.

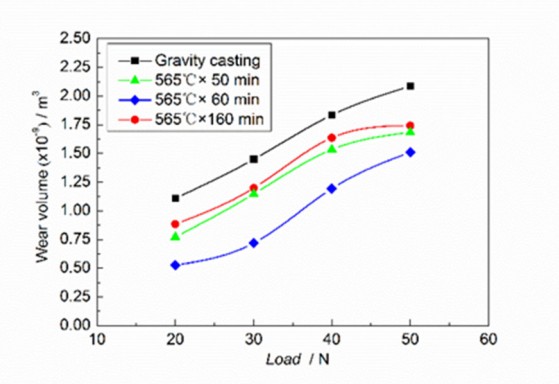

**Figure 4.** Curves of $Mg_2Si$–Al composites fabricated by using gravity casting and semi-solid extrusion with isothermal heat treatment holding times of 50 min, 60 min, and 160 min between wear volumes and applied load.

In addition, the curves also show that the wear resistances of $Mg_2Si$–Al composites fabricated by using semi-solid extrusion were better than those of $Mg_2Si$–Al composites fabricated by using gravity casting. Among them, the wear resistances of $Mg_2Si$–Al composite fabricated by using semi-solid extrusion with an isothermal heat treatment holding time of 60 min were best, followed by $Mg_2Si$–Al composites fabricated by using semi-solid extrusion with an isothermal heat treatment holding time of 50 min and 160 min. Compared with $Mg_2Si$–Al composites fabricated by using gravity casting, under high-load (50 N) conditions, the wear volumes of $Mg_2Si$–Al composites fabricated by using semi-solid extrusion with isothermal heat treatment holding times of 50 min, 60 min, and 160 min were 80.7%, 72.4%, and 83.5% of the former rates, respectively. Under the conditions of semi-solid extrusion, the $\alpha$-Al phase and eutectic structure were not only refined but their distribution became more uniform, and the spheroidization phenomenon of the $\alpha$-Al phase and $Mg_2Si$ appeared. This double spheroidizing effect reduced the cracking of the enhanced matrix and effectively improved the wear resistance of the composites.

Figure 5 shows the SEM and EDS results of the wear surfaces of $Mg_2Si$–Al composites fabricated by using gravity casting and semi-solid extrusion with 50 N applied load. It can be seen that the wear surface of the $Mg_2Si$–Al composite fabricated by using gravity casting was very rough; there were obviously deep furrows, fine cracks, and spalling on the wear surface, as shown in Figure 5a.

In contrast, only shallow furrows existed on the wear surfaces of $Mg_2Si$–Al composites fabricated by using semi-solid extrusion, and occasionally slight spalling was seen, as shown in Figure 5b–d. The furrows were the shallowest when the isothermal heat treatment holding time was 60 min, while the furrows were the deepest when the holding time was 160 min. At the same time, there were still some protruding $Mg_2Si$ particles and some accumulations on the wear surface. This was because the temperature of the wear surface was higher under severe wear conditions, which caused a strong plastic flow and wear surface. Some parts of the slab were piled up or plowed off to form abrasive dust, so the $Mg_2Si$ particles in the bottom layer were exposed. In addition, some $Mg_2Si$ particles fell off after being pulled out, and there were still marks remaining on the worn surface after their falling off. It can be inferred that severe wear generated a large amount of heat. This heat accumulation caused the local temperature of the wear surface to be very high, causing a flash temperature phenomenon and even melting. As a result, the bonding strength of the matrix and $Mg_2Si$ particles was reduced; $Mg_2Si$ particles were no longer constrained by the matrix to be fixed in the corresponding position, and finally the reinforced particles fell off from the matrix, forming a trace after falling off.

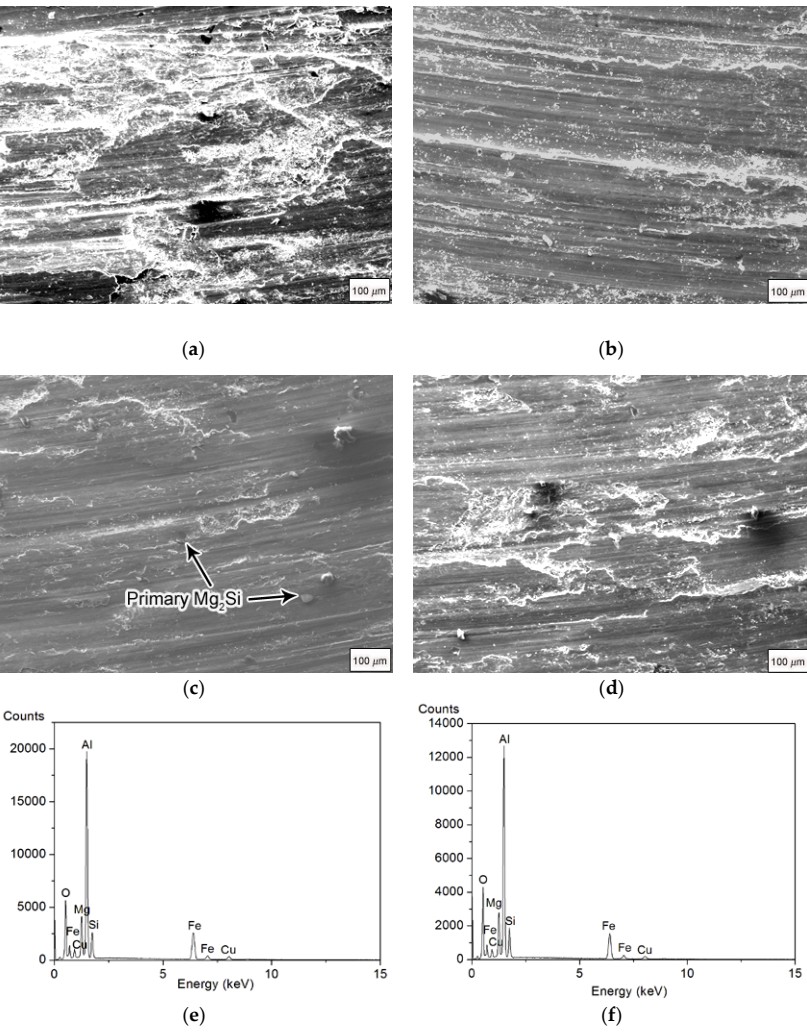

**Figure 5.** SEM and EDS results of the wear surfaces of $Mg_2Si$–Al composites with 50 N applied load. (**a**) Gravity casting, (**b**) semi-solid extrusion with isothermal heat treatment holding time of 50 min, (**c**) semi-solid extrusion with isothermal heat treatment holding time of 60 min, (**d**) semi-solid extrusion with isothermal heat treatment holding time of 160 min, (**e**) EDS result of $Mg_2Si$–Al composites fabricated by using gravity casting, and (**f**) EDS result of $Mg_2Si$–Al composites fabricated by using semi-solid extrusion.

Figure 5e,f show the EDS results of the wear surfaces of Mg$_2$Si–Al composites. The results showed that in addition to aluminum, magnesium, silicon, and copper, the wear surfaces of the two composites also contained iron and oxygen. The main component of the composite material did not contain iron. The iron came from a counter-mill disc made of 45 steel, indicating that iron element transfer occurred during the wear process. The presence of oxygen indicated the presence of Al$_2$O$_3$ on the worn surface. Qin et al. [20] found the wear debris was mainly composed of Mg$_2$Si, Al, and Al$_2$O$_3$ phases by XRD analysis of the abrasive debris of the Mg$_2$Si–Al composite. Generally, under the condition of mild wear (low load), the Al$_2$O$_3$ oxide film can protect the wear surface to a certain extent, delay the wear failure, and reduce the wear rate. However, during the severe wear stage, although an oxide film is produced, the surface is strongly softened due to the high load, severe wear, and increased temperature of the friction interface. At this time, the peeling rate of the surface layer is dominant, and the oxide film will be quickly worn away. Its formation rate may be much slower than the peeling rate of the surface layer, thereby limiting the protective effect of the oxide film. Therefore, even if the anti-wear effect of Mg$_2$Si particles is very prominent in the high-load stage, which makes the change of the wear curve slow, the wear volume of the composite material is still large.

Figure 6 shows the SEM image of abrasive debris from Mg$_2$Si–Al composite by using gravity casting and semi-solid extrusion with isothermal heat treatment holding time of 60 min with 50 N applied load. Under severe abrasion, the abrasive debris of the two composites consisted of large layers of lamellar debris and small particle agglomerates. Among them, the composition of layered debris was mainly matrix and Mg$_2$Si particles.

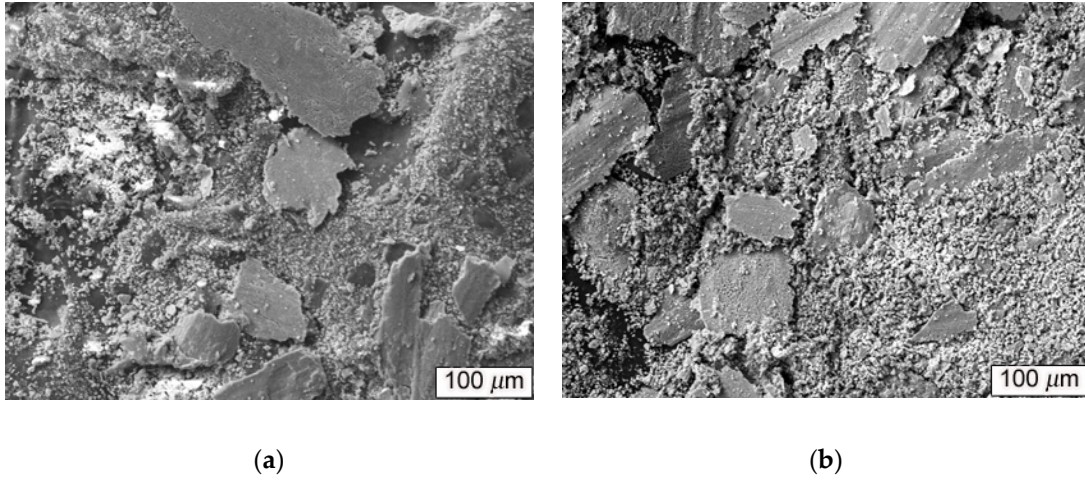

(**a**)                                              (**b**)

**Figure 6.** SEM of wear debris on Mg$_2$Si–Al composites fabricated by (**a**) gravity casting and (**b**) semi-solid extrusion with isothermal heat treatment holding time of 60 min, with 50 N applied load.

Figure 7 shows the wear curve of Mg$_2$Si–Al composites fabricated by using gravity casting and semi-solid extrusion with 30 N applied load. It can be seen that the wear volume of Mg$_2$Si–Al composites decreased with increasing sliding rate. The wear volume decreased faster between 300 and 700 r/min, and the decreasing trend slowed down between 700 and 1000 r/min. This was because between 300 and 700 r/min, the wear was a mild state of wear due to low load (30 N) and a slow sliding rate. Al$_2$O$_3$ on the wear surface played a certain protective role, and its comprehensive protection effect with work hardening was much greater than the peeling rate of the wear surface, coupled with the anti-wear effect of the Mg$_2$Si particles, so the wear volume was significantly reduced. In the range of 700–1000 r/min, a large amount of frictional heat was generated because of the high sliding rate, which resulted in a local high temperature of the friction interface and a reduced binding effect of the substrate on the surface and subsurface layers, although the load was still low. In addition, the increased surface temperature will also lead to a decrease in the shear strength of the subsurface layer [22], resulting in a strong softening of the worn surface, an increase in the peeling rate of the

surface layer, and an acceleration of the failure of the wear material, so the reduction trend of the wear volume becomes slow.

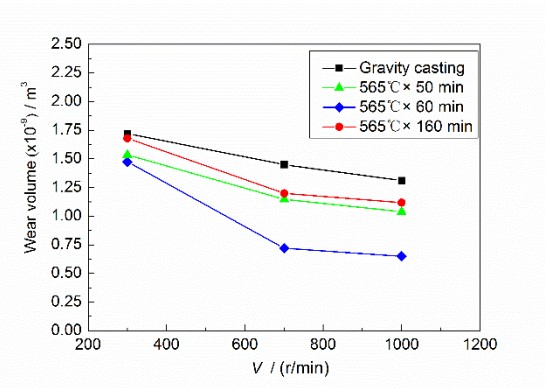

**Figure 7.** Curves of Mg$_2$Si–Al composites between wear volume and sliding velocity.

Figure 8 shows SEM of wear surfaces of Mg$_2$Si–Al composites fabricated by using gravity casting and semi-solid extrusion with a sliding velocity of 1000 r/min. The figure shows that plastic flow occurred on the wear surface of the Mg$_2$Si–Al composite, and craters and furrows remained after the plastic flow. The furrows on the wear surface of Mg$_2$Si–Al composite fabricated by using gravity casting were deep and wide (Figure 8a), while the furrows of Mg$_2$Si–Al composites fabricated by using semi-solid extrusion were shallow and thin (Figure 8b–d). When the holding times were 50 min and 160 min, the width and depth of the furrows of semi-solid extruded Mg$_2$Si–Al composites were not significantly different. When the holding time was 60 min, the furrows were the shallowest and hardly visible. In addition, prominent Mg$_2$Si particles, an accumulation phenomenon, and some peeling pits remaining after Mg$_2$Si fell off were still visible.

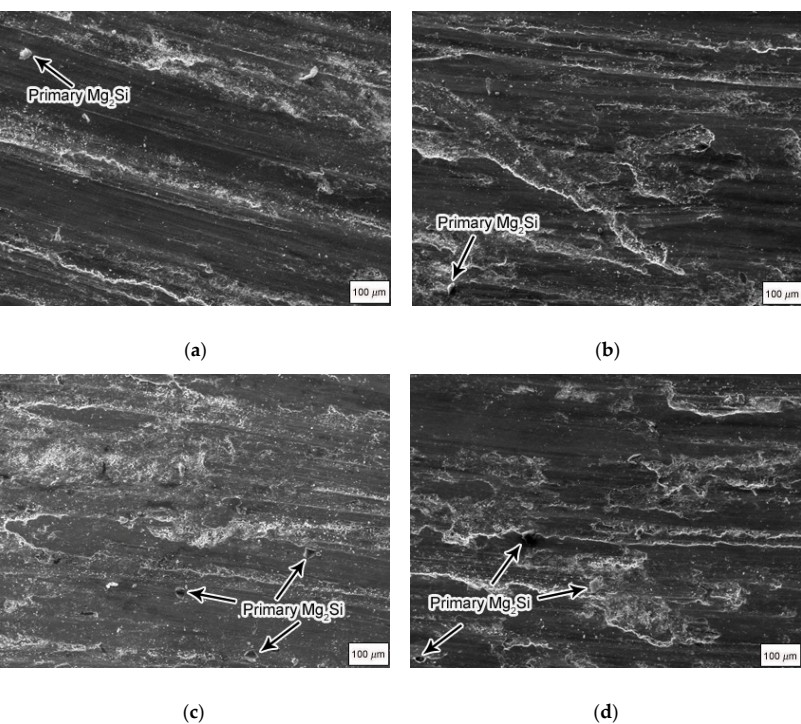

**Figure 8.** SEM of wear surfaces of Mg$_2$Si–Al composites fabricated by using (**a**) gravity casting and semi-solid extrusion with isothermal heat treatment holding times of (**b**) 50 min, (**c**) 60 min, and (**d**) 160 min with a sliding velocity of 1000 r/min.

Figure 9 shows the wear surfaces of $Mg_2Si$–Al composites fabricated by semi-solid extrusion at high magnification. It can be seen from the figure that $Mg_2Si$ reinforced particles and deposits existed on the worn surface. This was because the increase of the sliding rate promoted the plastic flow of the composite material. When the sliding rate was 1000 r/min, the friction surface generated heat and the temperature of the wear surface was high, resulting in a strong plastic flow. Some parts of the wear surfaces were pushed forward during the wear process, forming a pile, and at the same time, the $Mg_2Si$ reinforced particles in the bottom layer of this part were also exposed.

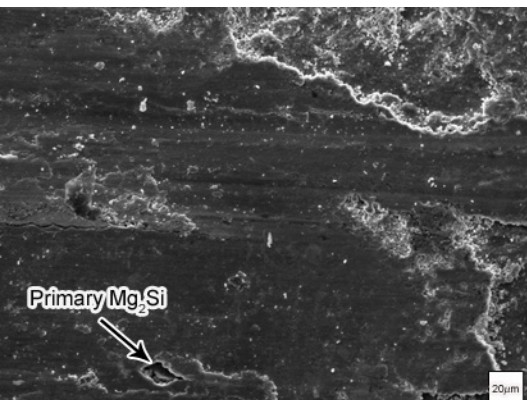

**Figure 9.** SEM of wear surfaces of $Mg_2Si$–Al composite fabricated by using semi-solid extrusion with isothermal heat treatment holding time of 160 min, with a sliding velocity of 1000 r/min.

Figure 10 shows the EDS result of wear surface of $Mg_2Si$–Al composite fabricated by using semi-solid extrusion with isothermal heat treatment holding time of 160 min, with sliding velocity of 1000 r/min. Similar to the EDS analysis of the wear surface under high load, iron and oxygen were also present on the worn surface, indicating that at a high sliding rate, the iron element in the grinding disc (45 steel) was transferred to the worn surface of the composite material, and there was also a protective layer of $Al_2O_3$ on the worn surface.

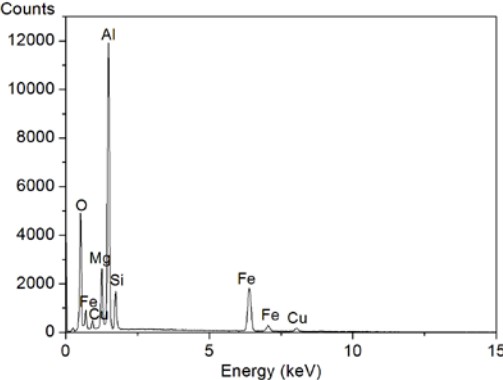

**Figure 10.** EDS result of wear surface of $Mg_2Si$–Al composite fabricated by using semi-solid extrusion with isothermal heat treatment holding time of 160 min, with a sliding velocity of 1000 r/min.

Kwork et al. [23] studied the friction behavior of $SiC_p$–Al composites at high speeds and found that the Al surface melted when the speed reached a certain value. Qin et al. [20] also believed that because of the influence of frictional heat, the worn surface locally reached a higher temperature and the surface melted, thereby reducing the binding effect of the Al matrix on the $Mg_2Si$ reinforced particles, causing the reinforced particles to fall off and a residual mark to form. In this experiment, the hard $Al_2O_3$ oxide film was continuously worn away and constantly produced. The oxide film and work hardening had a good protective effect on the composite material, which was a factor that

delayed the wear failure. However, as the sliding rate increased to 1000 r/min, the temperature of the wear surface increased, the shear strength of the alloy was reduced, and the plastic flow capacity of the surface layer and sub-surface layer was greatly increased, which accelerated the material failure rate. This was a factor contributing to wear failure. Under the condition of a low sliding rate, the favorable factors that delay the wear failure played a leading role, resulting in a rapid reduction in the volume of wear. With the increase of the sliding rate, the factors of accelerated wear failure played an increasingly important role. The combined effect of the two slowed down the trend of reducing the wear volume, and the characteristics shown in the wear curve appeared.

For $Mg_2Si$–Al composites, the dry sliding wear process is affected by both external and internal factors [24]. External factors include sliding distance, ambient temperature, form of motion, sliding rate, load, surface state, and grinding disc. Among many factors, load and sliding rate are two more important parameters, which have a greater impact on the wear failure process. The material of the grinding disc also has a certain effect on the dry sliding wear process. Since dry sliding friction and wear require a pair of friction pairs, the wear behavior of the grinding disc will also affect the wear resistance of the tested material. If the sample material is close to the microstructure of the disc material, then a conjugate friction pair will be formed during the pairing process due to the large mutual solubility of metals. This kind of friction pair has strong vibration and noise during the wear process, which results in severe wear of the tested materials and the grinding disc. Conversely, if the microstructure of the sample material is different from that of the disc material, the mutual solubility of the metal is small, resulting in a stable friction and wear process and less wear. In this experiment, 45 steel (composed of ferrite and pearlite) was used as the paired friction pair. Obviously, the microstructures of $Mg_2Si$–Al composites and the grinding disc are completely different, and the metal mutual solubility between the two is small; therefore, the material of the grinding disc has less influence on the wear behaviors of the composites.

Internal factors, such as the matrix and reinforcement (type, size, micromorphology, distribution, volume fraction, etc.) also affect the friction and wear behaviors of the composites. The matrix hardness of the composite material also greatly affects its wear resistance. According to the hardness test, the Brinell hardness of $Mg_2Si$–Al composite fabricated by using gravity casting was 115 HB, and the Brinell hardnesses of $Mg_2Si$–Al composite fabricated by using semi-solid extrusion with isothermal heat treatment holding times of 50 min, 60 min and 160 min were 153 HB, 160 HB, and 134 HB, respectively. Since the hardnesses of $Mg_2Si$–Al composites fabricated by using semi-solid extrusion were significantly improved, plastic flow on the worn surface did not easily occur, which reduced the wear volume of the wear sample, so their wear resistances were improved. Reinforcement particles also play a role in the wear process [24]. Studies have shown [25,26] that, for alloys, if there are fine hard particles in the matrix, their wear resistance will be correspondingly good. In general, harder particles have wear abrasion resistance [24,27]. The greater the volume fraction, the more uniform the distribution of the reinforcement, and the better its wear resistance. There is no obvious law on the effect of reinforcement size on wear resistance. In addition, during the wear process, stress concentration, micro-deformation, holes, and micro-cracks are more likely to breed at the interface of the reinforcement and the matrix due to the large difference in plastic deformation capacity between the reinforcement and the matrix, thereby reducing the properties of the composites. The microstructure directly affects the wear properties of alloys. Therefore, the micromorphology of the reinforcement has a greater impact on the wear behaviors. For $Mg_2Si$–Al composites fabricated by using semi-solid extrusion, the $Mg_2Si$ reinforced phase is spherical, small, and round, and the stress concentration around the particles is weak, so crack generation and propagation are delayed. Because the spherical $Mg_2Si$ particles have a higher bonding strength with the matrix, they do not easily fall off the matrix. Once the spherical $Mg_2Si$ particles come off the matrix, they only act as spherical abrasive particles, so the cutting and abrasion of the composite material is relatively small, forming a shallow furrow. In short, semi-solid extrusion changed the morphology of the primary $Mg_2Si$, which contributed significantly to the better wear resistance of the composite.

## 4. Conclusions

We undertook a comparative study of microstructures and wear behaviors of Mg$_2$Si–Al composites fabricated by using gravity casting and semi-solid extrusion. The main conclusions were as follows:

(1) After P modification, the primary Mg$_2$Si transformed to polygonal blocks. Extraction tests showed that the Mg$_2$Si crystals had octahedral and tetrakaidecahedral morphologies.

(2) The semi-solid microstructure of the double-spheroidized α-Al matrix and reinforced phase Mg$_2$Si was successfully obtained by using semi-solid extrusion. Extraction tests showed that the Mg$_2$Si crystals had a spherical morphology.

(3) The wear resistance of Mg$_2$Si–Al composites using different preparation methods was as follows: semi-solid extrusion with isothermal heat treatment holding time of 60 min > semi-solid extrusion with isothermal heat treatment holding time of 50 min > semi-solid extrusion with isothermal heat treatment holding time of 160 min > gravity casting, respectively. This was because the spherical and small Mg$_2$Si reinforced phase in Mg$_2$Si–Al composites fabricated by using semi-solid extrusion had a weaker stress concentration around them, so crack generation and propagation were delayed, and they did not easily fall off the matrix due to their higher bonding strength with the matrix. Once the spherical Mg$_2$Si particles came off the matrix, they only acted as spherical abrasive particles, so the cutting and abrasion of the composite material were relatively small.

(4) The wear failure mechanisms of Mg$_2$Si–Al composites fabricated by using gravity casting and semi-solid extrusion were mainly adhesive wear and abrasive wear.

**Author Contributions:** Conceptualization, X.L. and Y.Z.; Data curation, X.L.; Formal analysis, X.L., M.Y., and Y.Z.; Funding acquisition, X.L. and M.Y.; Investigation, X.L.; Methodology, X.L.; Project administration, X.L. and M.Y.; Resources, M.Y. and D.Z.; Supervision, D.Z.; Validation, X.L., M.Y., and D.Z.; Visualization, M.Y.; Writing—original draft, X.L.; Writing—review and editing, X.L. All authors have read and agreed to the published version of the manuscript.

**Funding:** This research was funded by the National Natural Science Foundation Regional Science Foundation Project of China (No. 51564005); the National Natural Science Foundation Youth Science Foundation Project of China (No.61901007); the Science and Technology Research Project of Education Department of Jilin Province, China (JJKH20180335KJ, JJKH20200040KJ); the Jilin Science and Technology Innovation Development Plan Project (No. 201750247); and the Research Projects Commissioned by Enterprises and Institutions of Beihua University (No. 201901015).

**Acknowledgments:** I would like to thank Yongbing Chen and Tianjun Bian from Jilin University for their help in the preparation of samples.

**Conflicts of Interest:** The authors declare that there is no conflict of interest regarding the publication of this article.

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
