# Peer review of "Microstructure and Wear Resistance of Mg2Si–Al Composites Fabricated Using Semi-Solid Extrusion"

_metals, doi:10.3390/met10050596_

Round 1
Reviewer 1 Report
In this paper the microstructure and the wear resistance of Mg2Si/Al composites fabricated using gravity casting and semi-solid extrusion were analyzed. Although the work is well conducted, the points indicated below need to be adequately addressed before the manuscript can be considered for publication.
- P3 L127 “The wear distance was 629 m under both variable loads and variable speed conditions.” I do not understand why a distance of 629 m was used. Please explain.
- How many specimens were tested in each condition?
- P6 L204 the authors say the sliding speed is 1000 rpm whereas in Table 1 it seems it is 700 rpm. Please revise.
- It is unclear what type of wear test was used. I guess was the pin-on-disk wear test, but it was not specified. It is unclear what the counter material and its geometry are.
- P12 L350 "For Mg2Si / Al compounds, the dry sliding wear process is influenced by both external and internal factors" How can the authors reach this statement? I wonder if the author used statistical tools (e.g. ANOVA) to evaluate the influence of the factors.
- There are some typos: P3 L120 “The sample size was φ5 mm × 12 mm.” The letter φ is Φ. P3 L126 “…and mw is the mass of the sample in distilled water” The symbol mw is m(subscript)w according with the equation. P3 L129 “Table 1. Experimental parameters for Test II and I.” Switch I with II. Please revise the whole manuscript.
With a revision addressing these points, the manuscript can be recommended for publication
Author Response
Response to Reviewer 1 Comments
Point 1: P3 L127 “The wear distance was 629 m under both variable loads and variable speed conditions.” I do not understand why a distance of 629 m was used. Please explain.
Response 1: It was the pin-on-disk wear test, carried out using an MG-200 type friction and wear tester with a disc material of 45 steel. Figure 1 shows the schematic of the contact method between the sample and the grinding disc. The value of wear distance S can be calculated according to the formula S = πDn, where D is the diameter, and n is the total number of revolutions. In this experiment, the diameter D is 23.847 mm, and the total number of revolutions is 8400 revolutions under both variable loads and variable speed conditions. Therefore, S = πDn=3.14×23.847×8400=628988 mm≈629 m.
|
Figure 1. Schematic of the contact method between the sample and the grinding disc.
Point 2: How many specimens were tested in each condition?
Response 2: Three specimens were tested in each condition, and the wear volume was determined from the average value of them. After revised, P3 L128 “…629 m under both variable loads and variable speed conditions. Three specimens were tested in each condition, and the wear volume was determined from the average value of them. ”
Point 3: P6 L204 the authors say the sliding speed is 1000 rpm whereas in Table 1 it seems it is 700 rpm. Please revise.
Response 3: P6 L204 the sliding speed is 700 rpm according to Table 1. After revised, the sentence is “ The friction and wear curves of Mg2Si/Al composites under different loads at a fixed sliding rate of 700 r/min are shown in Figure 4.”
Point 4: It is unclear what type of wear test was used. I guess was the pin-on-disk wear test, but it was not specified. It is unclear what the counter material and its geometry are.
Response 4: It was the pin-on-disk wear test. Figure 2 shows the schematic of dry sliding wear equipment. 45 steel (composed of ferrite and pearlite) was used as the paired friction pair, whose shape is disk-shaped, as shown in Figure 1.
P3 L120 “ …with a disc material of 45 steel whose shape is disk-shaped. It was the pin-on-disk wear test. ”
|
Figure 2. Schematic of dry sliding wear equipment.
Point 5: P12 L350 "For Mg2Si / Al compounds, the dry sliding wear process is influenced by both external and internal factors" How can the authors reach this statement? I wonder if the author used statistical tools (e.g. ANOVA) to evaluate the influence of the factors.
Response 5: In the current work, various influencing factors are complicated in the process of dry sliding wear. It is difficult to use statistical tools (e.g. ANOVA) to evaluate the influence of the factors. But we have consulted a lot of literature materials, summarized them, and used them to analyze the wear test results of this article. According to research [24][https://doi.org/10.1016/0043-1648(95)06657-8], the dry sliding wear process is affected by both external and internal factors, and reinforcement particles play a role in the wear process [24]. According to research [22][https://doi.org/10.1016/j.wear.2007.05.008], load and velocity have a great influence on dry sliding wear. According to researches [25][https://doi.org/10.1016/S0043-1648(01)00687-1] and [26][https://doi.org/10.1016/j.wear.2004.03.011], for alloys, if there are fine hard particles in the matrix, their wear resistance will be correspondingly good. According to researches[24][https://doi.org/10.1016/0043-1648(95)06657-8] and [27][https://doi.org/10.1007/BF02652716], harder particles have wear abrasion resistance. How to use statistical tools (e.g. ANOVA) to evaluate the influence of the factors is the next research goal of our research group. We will do further researches on this by doing more experiments, looking for rules, and analyzing the effects of influencing factors on wear behaviors.
Point 6: There are some typos: P3 L120 “The sample size was φ5 mm × 12 mm.” The letter φ is φ. P3 L126 “…and mw is the mass of the sample in distilled water” The symbol mw is m(subscript)w according with the equation. P3 L129 “Table 1. Experimental parameters for Test II and I.” Switch I with II. Please revise the whole manuscript.
Response 6: After revised, now P3 L120 “The sample size was φ5 mm × 12 mm.” P3 L126 “…and mw is the mass of the sample in distilled water”. P3 L129 “Table 1. Experimental parameters for Test I and II.”
Table 1. Experimental parameters for Test I and II.
Experiments |
Velocity (r/min) |
Load (N) |
Test I |
700 |
20 |
30 |
||
40 |
||
50 |
||
Test II |
300 |
30 |
700 |
||
1000 |

Reviewer 2 Report
Dear Authors,
Thanks for this nice piece of work. It merits publication in Metals, but with some more analyses I believe.
You indicate that Cu is partially present. This could create Guinier-Preston zones with the presence of Mg. As you know, such precipitated zones influence largely mechanical properties of aluminum. Prior works by Wintenberger should be read on this. But since your XRD diagrams are not fitted, one cannot estimate if the phase-check by Panalytical softwares are valid. You should use Rietveld analysis to estimate which phases are indeed present, and to quantify their amounts.
Unfortunately, you only give EDS spectra for most of the samples, which cannot ascertain the various phases. You could have also measured XRD diagrams or Raman spectra to identify and quantify all the phases in the composites.
So that I feel that before your work being publishable, a little more work is needed.
sincerely
Daniel Chateigner
Author Response
Response to Reviewer 2 Comments
Point 1:
Dear Authors,
Thanks for this nice piece of work. It merits publication in Metals, but with some more analyses I believe.
You indicate that Cu is partially present. This could create Guinier-Preston zones with the presence of Mg. As you know, such precipitated zones influence largely mechanical properties of aluminum. Prior works by Wintenberger should be read on this. But since your XRD diagrams are not fitted, one cannot estimate if the phase-check by Panalytical softwares are valid. You should use Rietveld analysis to estimate which phases are indeed present, and to quantify their amounts.
Unfortunately, you only give EDS spectra for most of the samples, which cannot ascertain the various phases. You could have also measured XRD diagrams or Raman spectra to identify and quantify all the phases in the composites.
So that I feel that before your work being publishable, a little more work is needed.
sincerely
Daniel Chateigner
Response 1: In the current work, copper was mainly added by adding Cu-14 wt.% P master alloy. The presence of Cu could create Guinier-Preston zones with the presence of Mg. Unfortunately, because this batch of test samples is incomplete, XRD diagrams or Raman spectra can not be measured to identify and quantify all the phases in the composites. However, the effects of the presence of copper were analyzed. The composition analysis we have done before showed that the copper content is around 4 wt.%. XRD results showed that CuAl2 was present, indicating that a part of Cu reacted with Al. Even if the remaining part of Cu formed Guinier-Preston zones with Mg, the number is relatively small. The small number of Guinier-Preston zones also had a very limited impact on wear resistance. In addition, Guinier-Preston zones influence mechanical properties of aluminum, but mainly the hardness and the strength. In the current work, the wear resistances of Mg2Si/Al composites with similar Cu content were studied. According to previous studies [18] [http://www.ams.org.cn/CN/10.3724/SP.J.1037.2013.00726 ]
and [22] [https://doi.org/10.1016/j.wear.2007.05.008], no precipitation zone was found to have a significant effect on wear resistance. Therefore, we think Guinier-Preston zones have little influence. The influence of Guinier-Preston zones is a very worthy research topic, and it is also our next research direction. In future research, we will use more detection methods and analysis methods to identify and quantify all phases in the composite material to obtain better research results.

Reviewer 3 Report
I consider the manuscript well written, the experimental details are carefully described and the results explained in detail. The introduction is giving the necessary background for the readers, while correspondence with literature/references completes the discussion chapters also.
Just a few comments:
I recommend the use of the same measure unit for hardness, it would facilitate results comparison and contribute to the unity of the manuscript (in Introduction the hardness of Mg2Si is reported in Nm-2, the elastic modulus in GPa and the authors results are shown in Brinell units.
Row 120 - the sign before the dimensions of the samples might be wrong, please check.
XRD figure hard to follow - could be enlarged.
Author Response
Response to Reviewer 3 Comments
Point 1: I recommend the use of the same measure unit for hardness, it would facilitate results comparison and contribute to the unity of the manuscript (in Introduction the hardness of Mg2Si is reported in Nm-2, the elastic modulus in GPa and the authors results are shown in Brinell units.
Response 1: In Introduction the hardness of Mg2Si is reported in Nm-2, which is the result of other people's research, to help readers understand what the author has explained. In the hardness test, the hardness value was determined from the average value of seven hardness readings on each sample with a hardness tester (Brinell Hardness Tester HB-3000B), using a 7350 N load and 30 s holding time. In order to be consistent with the test method, we think it is more appropriate to use Brinell as the unit.
Point 2: Row 120 - the sign before the dimensions of the samples might be wrong, please check.
Response 2: P3 L120 “The sample size was φ5 mm × 12 mm.” The letter φ is φ. After revised, now P3 L120 “The sample size was φ5 mm × 12 mm.”
Point 3: XRD figure hard to follow - could be enlarged.
Response 3: P5 L167 XRD figure has been enlarged.
|
Figure 2. XRD spectra of Mg2Si/Al composites.
